# Alpha-band phase modulates perceptual sensitivity by changing internal noise and sensory tuning

**April Pilipenko\*, Alexandra McGowan, Jason Samaha\***

Department of Psychology, University of California, Santa Cruz, Santa Cruz, United States

## eLife Assessment

This **important** study explores how the phase of neural oscillations in the alpha band affects visual perception, indicating that perceptual performance varies due to changes in sensory precision rather than decision bias. The evidence is **convincing** in its experimental design and analytical approach. This work should interest cognitive neuroscientists who study perception and decision-making.

**Abstract** Alpha-band neural oscillations (8–13 Hz) are theorized to phasically inhibit visual processing based, in part, on results showing that pre-stimulus alpha phase predicts detection (i.e., hit rates). However, recent failures to replicate and a lack of a mechanistic understanding regarding how alpha impacts detection have called this theory into question. We recorded EEG while six observers (6020 trials each) detected near-threshold Gabor targets embedded in noise. Using signal detection theory (SDT) and reverse correlation, we observed an effect of occipital and frontal pre-stimulus alpha phase on sensitivity (d'), not criterion. Hit and false alarm rates were counterphased, consistent with a reduction in internal noise during optimal alpha phases. Perceptual reports were also more consistent when two identical stimuli were presented during the optimal phase, suggesting a decrease in internal noise rather than signal amplification. Classification images revealed sharper spatial frequency and orientation tuning during the optimal alpha phase, implying that alpha phase shapes sensitivity by modulating sensory tuning towards relevant stimulus features.

## Introduction

Visually-sensitive rhythmic neural activity in the alpha-band (7–14 Hz) was among the first features ever discovered in the human electroencephalogram (EEG) over a century ago (**Berger, 1929**). The prominence of alpha activity in single trials of the raw occipital EEG implies that, at virtually any moment, millions of visual neurons are engaged in synchronous periodic activity. But what consequences does such large-scale activity have for human perception? Despite extensive research over the last century, much is still unknown about how ongoing alpha activity shapes visual perception (**Mushtaq et al., 2024**; **Pascucci et al., 2025**).

Two seminal papers, both published in 2009, shed new light on the perceptual consequences of alpha oscillations by showing that detection (i.e., hit rate) of brief near-threshold visual stimuli was predictable by the phase of pre-stimulus alpha oscillations (**Busch et al., 2009**; **Mathewson et al., 2009**). These findings gave rise to the now-popular idea that alpha oscillations impose phasic inhibition in the visual system such that each ~100 ms cycle reflects a transition between states of relative high and low neuronal excitability (**Mathewson et al., 2011**; **VanRullen, 2016**), perhaps even giving rise to temporal windows in perceptual processing (**Samaha and Romei, 2024**). Since 2009,

**\*For correspondence:**
apilipen@ucsc.edu (AP);
jsamaha@ucsc.edu (JS)

**Competing interest:** The authors declare that no competing interests exist.

a number of studies have observed similar effects of pre-stimulus alpha phase on visual detection (*Busch and VanRullen, 2010*; *Manasseh et al., 2013*; *Hanslmayr et al., 2013*; *Harris et al., 2018*; *Alexander et al., 2020*; *Zazio et al., 2022*). Several studies have also found that phosphene perception following near-threshold magnetic stimulation of early visual cortex is dependent on ongoing alpha phase (*Dugué et al., 2011*; *Samaha et al., 2017a*), highlighting the role of early visual areas. Indeed, a recent paper found that pre-stimulus alpha phase modulates the earliest afferent visual response measurable in the human evoked potential, suggesting an early, possibly thalamic contribution (*Dou et al., 2022*).

In tandem with the above positive findings, however, there have been many notable failures to directly and conceptually replicate phase effects on visual detection (*Chaumon and Busch, 2014*; *Benwell et al., 2017*; *Ruzzoli et al., 2019*; *Tseng et al., 2023*; *Melcón et al., 2024*; *Benwell et al., 2022*; *Sheldon and Mathewson, 2022*) or response times (*Vigué-Guix et al., 2022*; *Vigué-Guix and Soto-Faraco, 2023*). This combined with notable variability in certain features of the original results (e.g., some papers report alpha phase effects over occipital-parietal electrodes; *Mathewson et al., 2009*) while others find them over frontal electrodes (*Busch et al., 2009*) and variability in processing pipelines have prompted calls to both standardize methodological approaches as well as rethink the central idea that alpha imposes phasic inhibition in perception (*Keitel et al., 2022*).

Thus, a central question remains about whether pre-stimulus alpha phase modulates visual perception. However, even if such effects are reliable, previous approaches have lacked mechanistic insight into *how* alpha phase shapes perceptual processing. Virtually all the above-mentioned studies have analyzed changes in hit rates (i.e., detection rates of a stimulus that is always present) as a function of pre-stimulus phase, which leaves open the possibility that any effect is caused by an underlying change in sensitivity (d') or criterion (see *Figure 1C*). Distinguishing these accounts requires presenting target-absent trials at different alpha phases in addition to target-present trials, affording the computation of hit (HR) and false alarm (FAR) rates.

The current study aims to resolve these discrepancies by examining alpha-band phase effects in a near-threshold signal detection task (see *Figure 1A*) wherein both noise and target + noise trials are presented with equal probability in a large number of trials (6,020 per observer, n=6) across multiple EEG sessions. This approach ensures a sufficient number of trials in order to reliably compute signal detection theory (SDT) metrics across multiple alpha phase bins while also affording enough statistical power for reverse correlation analysis (*Xue et al., 2024*), making it preferred over having a larger sample size with fewer trials. If alpha phase indeed modulates perceptual sensitivity (d'), as opposed to criterion, we theorized that a phasic modulation of d' could be explained by one of two accounts, either a multiplicative gain or variance reduction model (see *Figure 1C*). The gain model suggests that sensory input is amplified by a phase-dependent multiplicative factor, effectively increasing the signal strength during optimal phases. This model predicts increased HR at optimal phases and proportionally smaller increases in FAR leading to a boost in d'. In contrast, the variance reduction model posits that optimal phases are associated with decreased trial-by-trial variability in both the signal and noise distributions. This account also predicts increased HR at optimal phases but, critically, predicts a concurrent *decrease* in FAR due to the narrower noise distribution. Finally, by collecting a large number of trials per observer with stochastic filtered noise in our stimulus and repeated stimulus sequences (i.e., double-pass), we used reverse correlation techniques and response consistency metrics (*Xue et al., 2024*) to examine how alpha phase changes sensory tuning and internal noise, respectively.

## Results

### Experimental paradigm and performance

Each participant (n=6) completed 5–6 EEG sessions of a yes/no detection paradigm whereby participants reported the presence or absence of a brief (8 ms; one screen refresh) vertical Gabor target (two cycles per degree) with concurrent confidence judgments (see *Figure 1A*), along with an additional imagination judgment (reported in the supplemental materials). Each trial contained either the target embedded in filtered noise (50% of trials) or noise only, presented to the right or left of fixation with random probability (see 'Methods'). The contrast of the target was titrated across blocks (43 blocks)

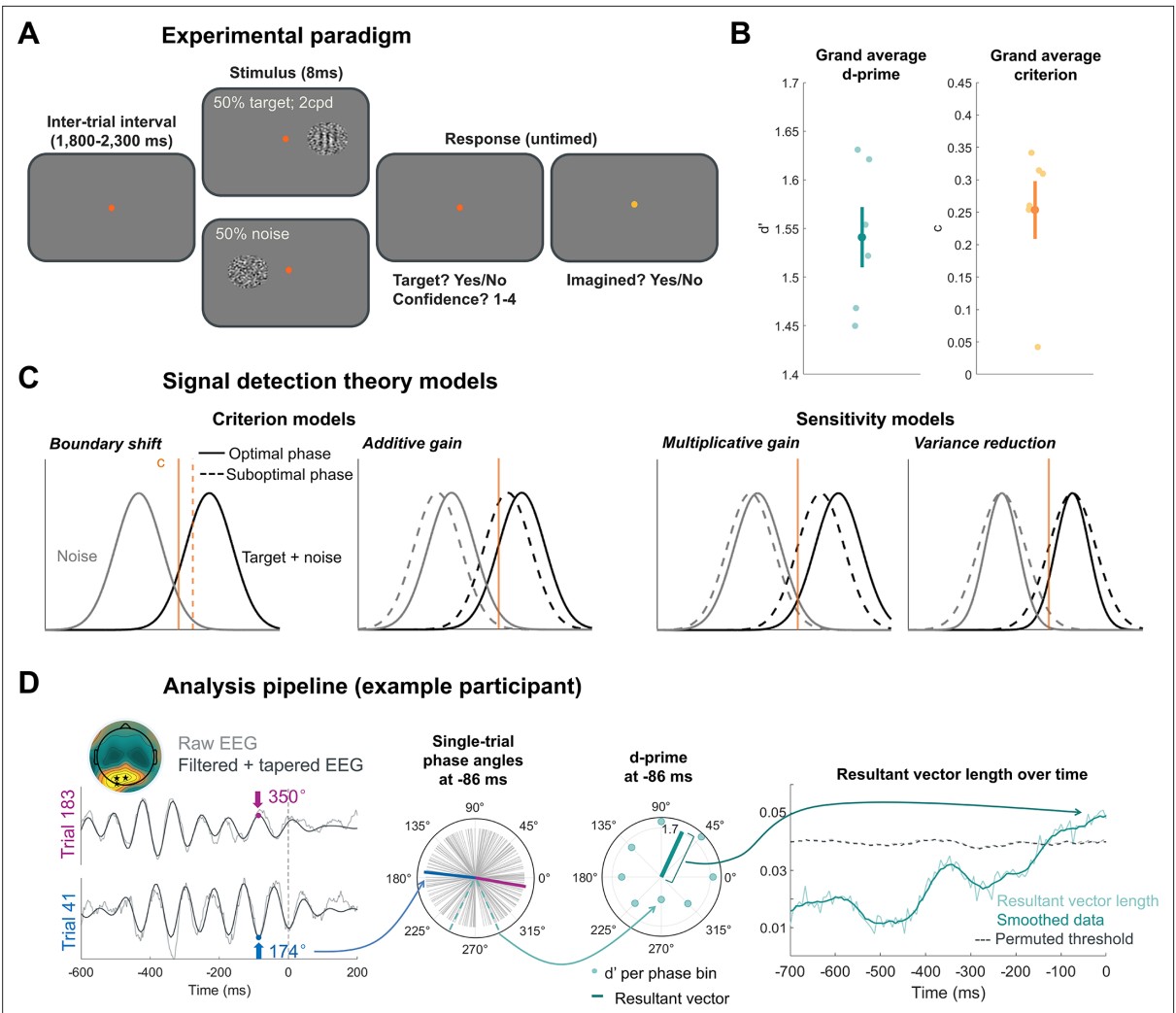

**Figure 1.** Experimental paradigm and analysis pipeline. (**A**) Task design. Each trial contained a brief, filtered-noise stimulus (8 ms) presented to the right or left of fixation with equal probability. Half of the trials had a target Gabor (two cycles per degree, 0° orientation) embedded in the noise (stimulus present), and the other half were only noise (target absent). Participants reported the presence of the target with concurrent confidence judgments followed by a question probing the possibility that what they had seen (either target or just noise) had been imagined (see supplemental materials). (**B**) Task performance (n=6). The contrast level of the target was titrated across task blocks to maintain a d' value between 1.2 and 1.8. The individual subject d' and criterion values are plotted in lighter circles; the grand average d' (1.54) and criterion (0.25) are the darker circles with error bars indicating ±1 standard error of the mean (SEM). (**C**) Phase effects on criterion and/or sensitivity in the signal detection theory (SDT) framework. Criterion and sensitivity changes may arise from various underlying mechanisms. A criterion change is indicative of a concurrent increase (or decrease) in hits and false alarms, with no change in separability between distributions. The first model (from left to right) indicates a phase effect on criterion as a result of a change in the internal decision criterion, whereas the second model shows a criterion effect from an additive gain applied to both the noise and signal + noise distributions (note that these two different criterion mechanisms would produce identical changes in behavior). A sensitivity shift induces a disproportionate change in hits compared to false alarms, leading to a change in overlap between distributions. The third model demonstrates multiplicative gain, whereby each distribution is multiplied by a small constant leading to a relatively small increase in the noise distribution (more false alarms) and a greater increase in the signal + noise distribution (even more hits). The final model shows the variance reduction account in which the variability of both distributions decreases, leading to more hits and fewer false alarms during the optimal phase. (**D**) The analysis pipeline illustrated for a single example participant (n=1). To quantify the coupling between alpha phase and behavior (in this example, d'), we computed the instantaneous phase using a Hilbert transform on the filtered (IAF ±2 Hz) and (post-stimulus) tapered EEG data. For each time point and electrode, the single-trial phase angles were binned into eight phase bins, and d' was computed using those subsets of trials. These measurements were then vectorized by assigning d' as the vector length and the bin center as the vector angle. The resultant vector length was then computed to quantify the alpha phase-d' coupling magnitude at each point in time while the angle points in the direction of the optimal phase. This example subject showed significant phase-d' coupling in the ~100 ms preceding stimulus onset.

to maintain performance at a d' between 1.2 and 1.8, leading to a grand average d' of 1.54 (SD: 0.08) and criterion of 0.25 (SD: 0.11), as seen in *Figure 1B*.

## Alpha phase-d' coupling supports phase-dependent noise reduction

To quantify the relationship between alpha phase and behavioral performance (d' and c), we first computed the instantaneous phase using the Hilbert transform on single-trial, filtered, and tapered EEG data. To prevent post-stimulus activity from contaminating the pre-stimulus window during filtering while also maximizing the amount of data for peri-stimulus phase estimation, we applied a temporal taper to the raw EEG signal, starting at +30 ms and linearly reducing the signal to zero µV by +70 ms (before any stimulus-evoked responses). We then applied a non-causal bandpass filter centered on each participant's individual alpha frequency (IAF ± 2 Hz) followed by a Hilbert transform to extract phase angles (*Figure 1D*). At each time point and channel, single-trial phase angles were binned into eight equally spaced bins centered at 0°, 45°, 90°, 135°, 180°, 225°, 270°, and 315° (bin width: ±22.5° relative to each bin center). For each bin, we calculated sensitivity and criterion, then represented these values as vectors whereby the angle indicated the phase bin center, and the length corresponded to performance magnitude (either the d' or c measurement). The resultant vector (i.e., the sum of the vectors across bins) indexed both the preferred phase (angle) and the degree of phase-dependent modulation (length). Under the null hypothesis of no phase dependence, vector length approaches zero. In contrast, a significant phase effect produces a longer vector with the angle indicating the optimal phase. After smoothing the resultant vector length over time with a 50 ms moving average, we compared the observed vector lengths to a permuted threshold (95th percentile of 1000 permutations) at each time point from –700–0 ms and performed cluster correction (95th percentile of the permuted cluster size) to account for multiple comparisons. All individual participant data can be found in *Figure 2–figure supplement1*.

The data revealed significant cluster-corrected coupling between alpha phase and d' in the prestimulus window from –220 ms until stimulus onset (cluster p=0.046), along with an additional coupling effect between –426 and –320 ms that did not survive cluster correction. The peak resultant vector length of around 0.034 corresponds to a peak-to-trough d' modulation of around 0.21 (range across participants: 0.18–0.33 SD units). No significant relationship was observed with criterion at any time point (see *Figure 1C*), suggesting that while sensitivity is modulated in a phase-dependent manner, the amount of internal evidence needed in favor of one decision or the other is relatively stable across the alpha cycle. We also found that pre-stimulus phase significantly predicted confidence ratings in a similar window as the d' effect, but this effect did not survive cluster correction (see *Figure 2–figure supplement2*). On the other hand, pre-stimulus alpha *power* was only predictive of confidence, not d' or criterion (see *Figure 2–figure supplement 3*), consistent with work linking alpha power to subjective perceptual reports (*Samaha et al., 2017b*; *Benwell et al., 2017*; *Samaha et al., 2020*; *Samaha et al., 2022*; *Benwell et al., 2022*).

To interrogate the robustness of our findings at the single-subject level, we adopted a test of binomial probability, which is a statistical framework that treats each individual as an independent replication and is ideal for small sample size studies that utilize a large number of trials per observer (*Schwarzkopf and Huang, 2024*). For our data, we assessed individual significance by averaging the actual and permuted resultant vector lengths across time (–450–0 ms) and comparing the real vector length to the 95% percentile of the permuted datasets. With this approach, three out of six participants showed significant d'-phase coupling, which corresponds to a binomial probability of p=0.002, indicating a very low probability that we observed these results by chance alone.

Our next goal was to investigate whether phase-dependent d' modulations were best characterized by a multiplicative gain or variance reduction model. The exact phase where performance is highest varies between participants due to cortical anatomy and retino-geniculate-striate conductance delays (*Busch and VanRullen, 2010*; *Busch et al., 2009*; *Alexander et al., 2020*), requiring us to phase-align the performance data across participants in order to visualize the underlying phasic effects. To this end, we aligned all metrics (d', c, HR, and FAR) by circularly shifting the data so that the bin with the highest d' was assigned to bin 8, which was then omitted from further visualizations. As seen in *Figure 2*, the underlying pattern of d' follows a phasic trajectory that waxes and wanes across the cycle, whereas criterion remains relatively stable. Importantly, the HR and FAR show an inverse phasic relationship, with FAR increasing as a function of the distance from the best phase bin and HR

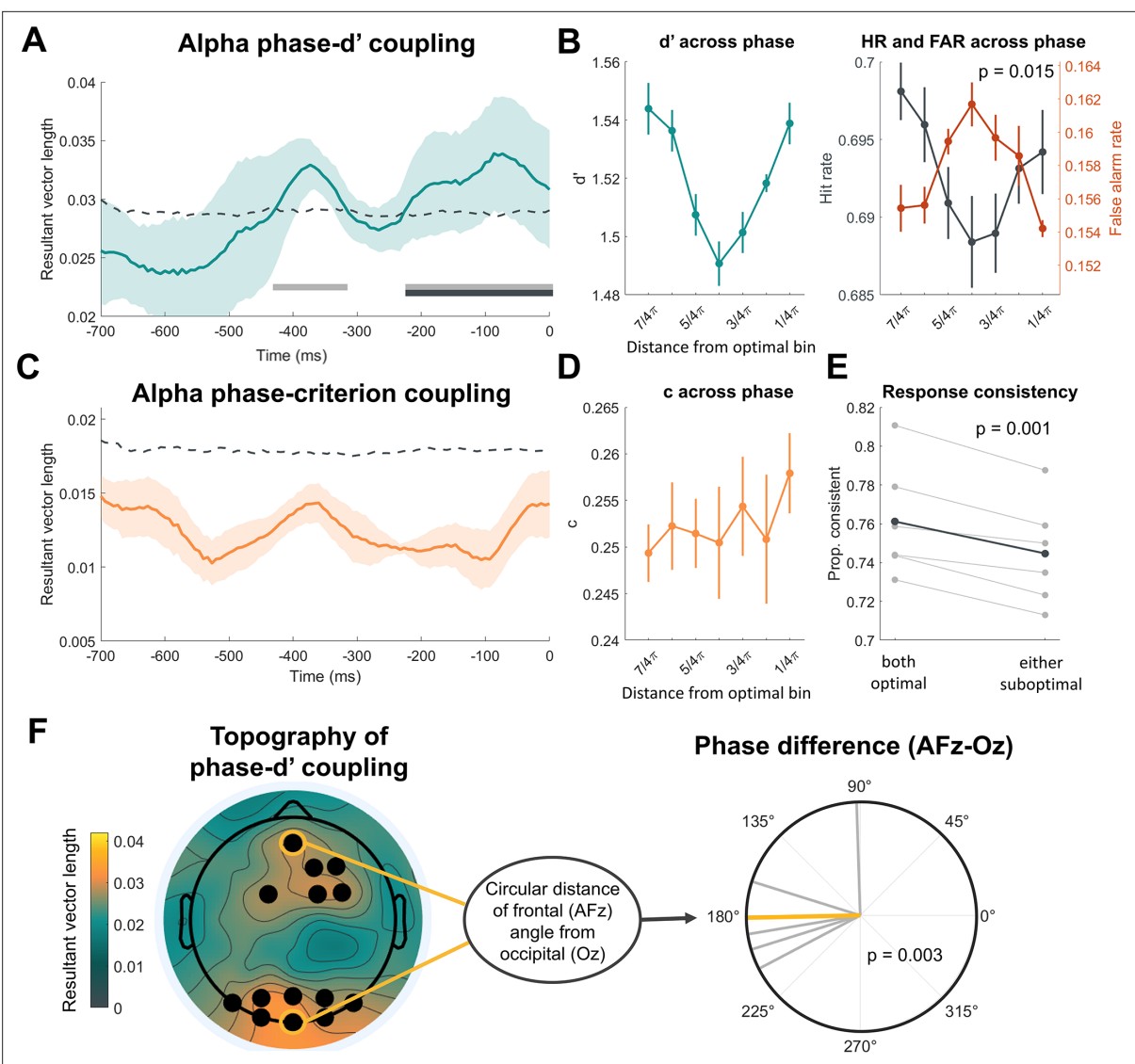

**Figure 2.** Alpha phase modulates perceptual sensitivity through phase-dependent fluctuations in internal noise (n=6). (**A**) Alpha phase predicts sensitivity up to 426 ms before the stimulus presentation. The time course of the resultant vector lengths characterizing the alpha phase-d' coupling shows significant cluster-corrected coupling from –220 ms until stimulus onset, as indicated by the black bar. A significant (uncorrected) effect was present as early as –426 ms pre-stimulus, as seen by the gray bar. The dashed line indicates the permutation threshold of the 95th percentile, and the error bars represent ± 1 SEM. (**B**) The phasic modulation of d' arose from the inverse phasic relationship of hits and false alarms, supporting a model in which the variability of internal responses changes in a phase-dependent manner. The first graph (from left to right) shows the underlying change in d' across phase bins. The data have been aligned across participants so that each individual's highest d' was assigned to bin 8 (omitted from the plot), with the remaining data circularly shifted, and is averaged across –450 ms to stimulus onset. This graph is for visualization purposes only. Error bars represent ± 1 SEM. The pattern shows a clear phasic modulation of d' across bins. The second graph shows the similarly-aligned and averaged hit rate (HR) and false alarm rate (FAR). A circular v-test of the resultant vector angles revealed an inverse phasic relationship between HR and FAR, consistent with the prediction made by the variance reduction model. (**C**) No evidence for the coupling of criterion to pre-stimulus alpha-band phase. Graph C reveals the time course of the resultant vector lengths for alpha phase-criterion coupling, which shows no significant phase-dependent relationship between phase and criterion. (**D**) The underlying shifted c across phase bins (shifted to participants' optimal phase, as in graph B) did not visually demonstrate a phasic modulation pattern. (**E**) Responses to pairs of identical stimuli occurring over the course of the experiment were more consistent with one another when both stimuli were presented during the best phase. This supports the notion that the internal representations of the stimulus were less variable during one's optimal phase. Error bars represent ±1 within-subjects SEM. (**F**) The topography of phase-d' coupling averaged over –450–0 ms showed significant effects (black dots) over frontal and occipital electrodes. The angular difference between the frontal (AFz) and occipital (Oz) optimal phases suggests a single dipole source as they are significantly clustered around ~180° (i.e., phase opposition). Gray lines are individual subject phase difference vectors and the orange line is the circular average.

The online version of this article includes the following figure supplement(s) for figure 2:

*Figure 2 continued on next page*

*Figure 2 continued*

**Figure supplement 1.** Individual participant graphs (n=1).

**Figure supplement 2.** Alpha phase shows a weak relationship with confidence and no relationship with judgments of imagination (n=6).

**Figure supplement 3.** Alpha power only shows a modulation of confidence (n=6).

decreasing. This pattern is consistent with the variance reduction model, but not the multiplicative gain account (see *Figure 1C*). We tested for phase opposition by taking the circular distance between the HR and FAR resultant vector angle and testing whether these values were significantly clustered around a mean of 180° (complete opposition) using a circular V-test. Indeed, this showed that the difference between the HR and FAR vector angle was significantly clustered around a mean of 180° (v=3.78, p=0.015), indicating that the phase angle associated with the greatest hits was counterphase to the phase angle associated with the greatest false alarms.

If alpha phase impacts perceptual sensitivity through a reduction in the variability of internal responses, we also expected that, in cases where participants were presented with identical stimuli, they would respond more consistently when the two stimuli had both been presented during the participant's optimal phase (i.e., the resultant vector angle of a given time point ± 90°). To examine this, we implemented a double-pass stimulus presentation (~600 stimulus pairs for participants 1–3 and ~2500 pairs for participants 4–6; see 'Methods') and analyzed participants' response consistency (*Xue et al., 2024*) to two identical stimuli. We found that participants were significantly more consistent with their detection reports when both stimuli were presented during their optimal phase compared to when either occurred during a suboptimal phase (t(5) = 6.60, p=0.001, 95% CI: [0.01, 0.02], *Figure 2E*). This evidence further corroborates the variance reduction account of alpha phase.

Lastly, by computing the phase-d' resultant vector length at every EEG channel, we found significant effects clustered both over frontal and occipital regions (see *Figure 2F*). This topography is consistent with prior studies that found phase effects in either one or the other region (frontal: *Busch et al., 2009*; *Hanslmayr et al., 2013*; occipital: *Mathewson et al., 2009*; *Spaak et al., 2014*; *Alexander et al., 2020*). We reasoned that these seemingly distinct spatial effects may actually reflect the same underlying alpha source just from opposite ends of a single dipole. If this were the case, then we would expect the resultant vector angle of a given time point to be counterphase, with a difference of 180°, between the frontal and occipital channels. A V-test of the circular distance between the resultant vector angle of channel AFz and Oz showed that the phase difference was significantly clustered around a mean of 180° (v=4.70, p=0.003), supporting the interpretation that previously reported alpha phase modulations in frontal and occipital regions both arise from the same alpha mechanism just with opposing phases.

So far, our results indicate a clear phasic modulation of d' across the alpha cycle with no accompanying change in criterion, as well as a tendency for participants to respond more consistently to two identical stimuli when they were both presented during their optimal phase. The underlying change in the FAR, which decreased during one's optimal phase, implies a reduction in the variance of the internal distributions as opposed to multiplicative gain.

## Alpha phase impacts sensory tuning of stimulus-relevant features

Our final goal in investigating phase-dependent sensitivity shifts was to determine how sensory tuning for key stimulus features (specifically, orientation and spatial frequency) varies as a function of alpha phase. To evaluate the tuning profiles, we applied reverse correlation to generate classification images (CIs), which reflect sensitivity to different stimulus features driving participants' target detection responses (see 'Methods'). This involved convolving each trial's stimulus with a pool of Gabor filters in quadrature phase that spanned a range of spatial frequencies (0.5–4 cycles per degree [cpd]) and orientations (−80° to 80°) centered around the target parameters (2 cpd, 0°), quantifying the energy present for each feature combination in each stimulus. These energy profiles were normalized across trials, then regressed on participants' binary responses (target present = 1; noise only = 0) to yield beta values. The resulting beta weights for each spatial frequency and orientation combination formed the CI (see *Figure 3A*). In order to assess changes in the sensory tuning, we analyzed the modulation of three parameters from a 2-D Gaussian fit to the CIs derived from trials where the stimulus occurred during the optimal and suboptimal phases using a bootstrapped analysis. The first

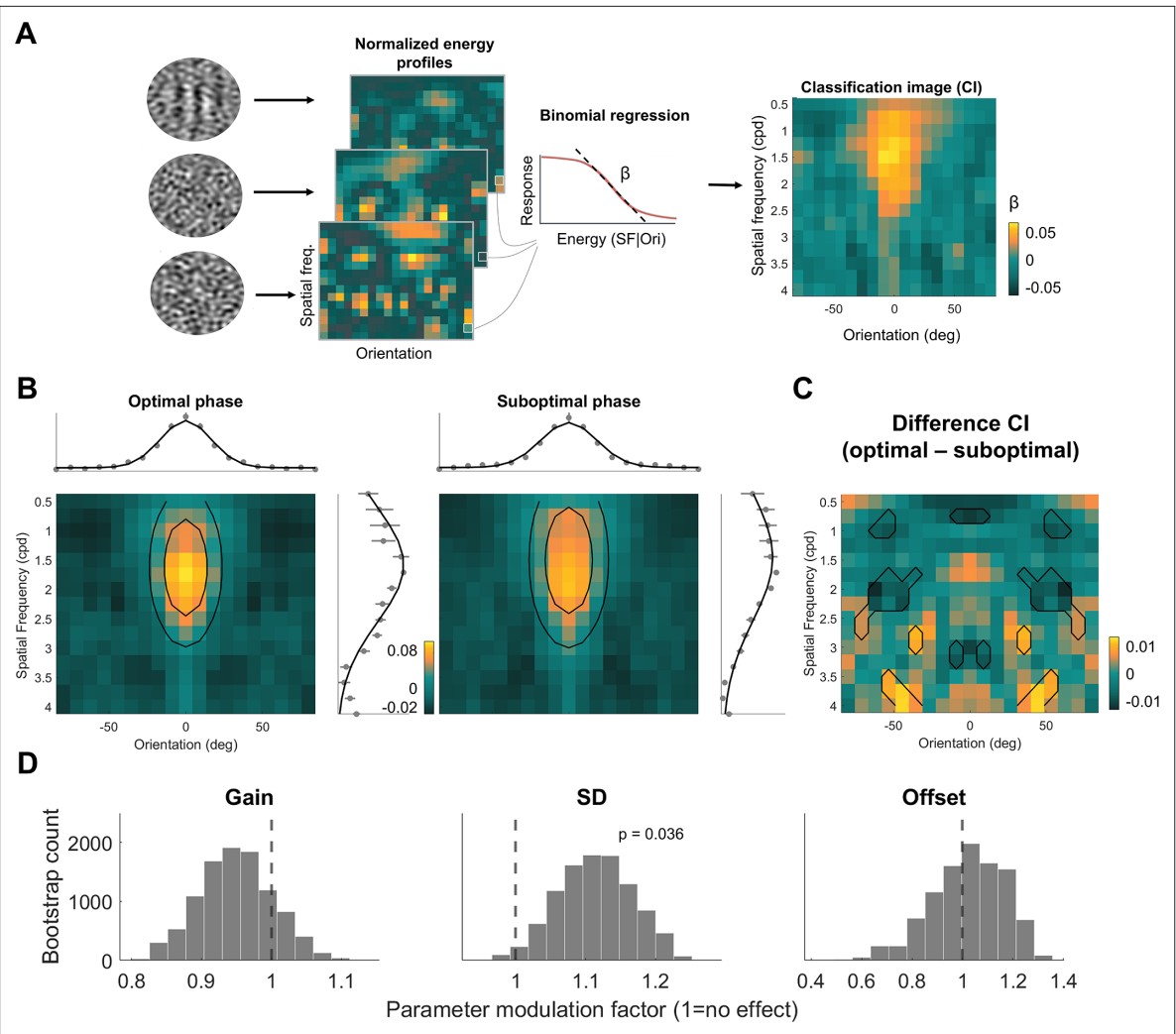

**Figure 3.** Alpha phase sharpens sensory tuning during the optimal phase (n=6). (**A**) The analysis pipeline for a single example participant (n=1). To create the classification images (CI) for each participant's optimal and suboptimal phase, we extracted the normalized energy profiles of each stimulus, quantifying the amount of energy in each orientation and spatial frequency (SF) combination. The single-trial energy at each SF and orientation was then regressed on the participant's binary response (i.e., 'target' or 'noise') resulting in a beta value. The beta values of a given SF and orientation characterize the influence that energy changes have on a participant's response and reveal perceptual tuning. (**B**) CI of participant's optimal and suboptimal phase. The bootstrapped CI of optimal and suboptimal phase with two example contours of the 2-D Gaussian fit depicted in black circles. The marginals of the CI averaged across orientations (above) and SF (on the right) are shown alongside each CI with the model fit. Error bars indicate 1 bootstrapped SEM. (**C**) The difference CI reveals broader tuning for the suboptimal phase around the target features. The difference CI was created by subtracting the optimal CI from the suboptimal CI for each bootstrapped iteration. The black outlines represent SF and orientation combinations that show significant (p<0.05) changes in beta values between the optimal and suboptimal phase. The differences reveal greater beta values for off-target features (target SF = 2, orientation = 0°) during one's suboptimal phase, broadening perceptual tuning. (**D**) Variance in sensory tuning is reduced during one's optimal phase. To quantify changes in sensory tuning between optimal and suboptimal phase, we fit a 2-D Gaussian to the optimal phase CI and modulated three parameters to account for the fit of the suboptimal phase CI. We found a significant modulation in the bootstrapped SD parameter, characterizing the variance of the Gaussian across both the SF and orientation dimensions. No significant modulation for the gain or offset parameters were found. Taken together, this suggests that sensory tuning during one's optimal phase sharpens towards the relevant stimulus features.

parameter modulates the gain of the 2-D Gaussian, reflecting the peakedness or sensitivity towards the target SF and orientation; the second parameter is the standard deviation (SD) of the Gaussian, representing the width of the tuning across both SF and orientation; and the third parameter estimated the offset of the distribution, capturing any general bias to all SF and orientations (akin to a criterion shift). We initially fit the CI from the optimal phase trials (mean r2 of the optimal phase = 0.87), and then used those parameters to fit suboptimal phase CI, while allowing gain, offset, and SD to vary by their own separate modulation factors (mean r2 of the suboptimal CI = 0.88; see *Figure 3B*). The

center of the 2-D Gaussian also varied but was not of interest for hypothesis testing. Thus, a modulation factor of 1 indicates no change in the parameter between optimal and suboptimal phase, whereas a modulation factor greater than 1 indicates that an increase in that parameter is needed in order to explain the suboptimal CI data.

Bootstrapping revealed a significant increase in the SD parameter in the suboptimal versus optimal phase CIs (p=0.036) but no significant modulation in either the gain (p=0.339) or offset (p=0.831) parameter (see *Figure 3D*). This finding suggests that the reason the suboptimal alpha phase is associated with increased internal noise is because of broader sensory tuning towards off-target stimulus features. To further interrogate this difference, we subtracted the suboptimal and optimal phase CIs for each bootstrap to reveal the SF and orientation weight differences between phases (see *Figure 3C*). There were significant differences (p<0.05) in a variety of orientations and spatial frequencies surrounding the target which indicated a greater weighting of these off-target features during the suboptimal phase. Additionally, we found some small but paradoxical higher weightings in the higher SFs for the optimal phase. To ensure that any differences in the CIs between optimal and suboptimal phases were not driven by difference in the overall proportion of 'target present' responses, we tested for differences in target present responses using a two-tailed paired-samples t-test. This analysis showed no significant difference in the proportion of 'target present' responses between the two phases (t(5) = –0.28, p=0.788, 95% CI: [–0.01, 0.01]) despite a significant difference in d' (t(5) = 10.59, p<0.001, 95% CI: [0.09, 0.15]). This finding is consistent with the variance reduction model since the increase in HR during one's optimal phase ($M_{optimal}$ = 0.71, $SD_{optimal}$ = 0.04; $M_{suboptimal}$ = 0.69, $SD_{suboptimal}$ = 0.03) is accompanied by a concurrent decrease in the FAR ($M_{optimal}$ = 0.15, $SD_{optimal}$ = 0.03; $M_{suboptimal}$ = 0.16, $SD_{suboptimal}$ = 0.03) leading to an approximately equal proportion of 'target present' responses ($M_{optimal}$ = 0.43, $SD_{optimal}$ = 0.03; $M_{suboptimal}$ = 0.43, $SD_{suboptimal}$ = 0.03).

In sum, our reverse correlation analysis revealed a modest enhancement in sensory tuning towards relevant stimulus features during the participant's optimal phase. This enhancement arose from considering a more precise range of SF and orientations relative to the target features as opposed to a boost in sensitivity of the target features. Taken together, our results support a functional role of spontaneous alpha in phasically enhancing perception via internal noise reduction and a sharpening of sensory tuning.

## Discussion

We investigated the role of spontaneous alpha-band phase in visual perception using a yes/no detection paradigm with concurrent human EEG recordings. In contrast to several recent null effects, our findings reveal a robust phasic modulation of perceptual sensitivity, without corresponding changes in decision criterion. This suggests that the phase of ongoing alpha oscillations corresponds to fluctuations in the sensitivity with which visual stimuli are perceived. These results align with prior evidence of phase-dependent changes in hit rate (*Busch et al., 2009*; *Mathewson et al., 2009*; *Dugué et al., 2011*; *Hanslmayr et al., 2013*; *Spaak et al., 2014*; *Alexander et al., 2020*) and extend this body of work by demonstrating that alpha phase modulates perceptual sensitivity within a signal detection theory framework. Moreover, our data are consistent with a model in which the variability of internal responses changes systematically across the alpha cycle, as reflected in the inverse relationship between hit rate and false alarm rate. This suggests that during one's suboptimal phase there is an increase in the overall variability of neural responses, leading to noisier sensory representation and, consequently, a less consistent perception of an identical stimulus. Together with decreases in the variance of sensory tuning during the optimal phase, our results suggest that alpha phase impacts sensitivity by shaping trial-to-trial variation in internal noise during perceptual decision making, leading to better matches between sensory evidence and decision templates as opposed to a change in the gain of internal sensory responses.

Two prior studies examining alpha phase within the signal detection theory (SDT) framework have yielded mixed results, with one study reporting no effect of phase on sensitivity or criterion (*Tseng et al., 2023*), and another finding a modulation of criterion (*Sherman et al., 2016*). *Tseng et al., 2023* replicated a phasic modulation of HR but observed no significant changes in FAR, d', or criterion. One possible explanation is their use of an online phase-estimation protocol that delivered stimuli at fixed phase angles (0°, 90°, 180°, 270°) and compared responses between 90° and 180°. This method does not account for individual differences in retinal delay (i.e., the time required for visual signals to reach

early visual cortex) and may have failed to target participants' truly optimal and suboptimal phases, thereby reducing sensitivity to phase effects (see: *Alexander et al., 2020*). In contrast to Tseng et al. and our own findings, *Sherman et al., 2016* reported that alpha phase modulates criterion. Notably, this effect was contingent on stimulus expectation and implied that there is a specific phase most likely to be biased by prediction. A key difference in their design was the explicit manipulation of both attention and expectation, which are factors that were randomized in our study. This raises the possibility that spontaneous alpha activity primarily affects d', whereas expectation-driven modulations of alpha reflect top-down processes that influence criterion.

We also note that the observed modulation of d' between optimal and suboptimal phases was relatively modest in absolute terms (0.21) in our study and could therefore require many trials per subject to detect. Two reasons for this modest effect size could be related to specific features of our task design. First, we titrated stimulus contrast to maintain consistent task performance. This titration could have reduced the magnitude of the phase effect on d' that would otherwise be apparent if the stimulus intensity were kept constant. Additionally, the use of (relatively) high-contrast random noise likely means that trial-to-trial variability in perception is largely driven by random fluctuations in the noise properties and, to a lesser extent, internal brain state. Although both of these choices were necessary to perform SDT and reverse correlation analysis, they differ from many previous studies investigating alpha phase using only near-threshold detection in the absence of external noise and may contribute to an underestimation of the true effect size.

By using reverse correlation we discovered reductions in the sensory tuning width (as measured through SD) during the optimal phase of pre-stimulus alpha. The finding of broader tuning during suboptimal phases supports the notion that internal representations are noisier and more variable in this state. This finding provides mechanistic insight into how the phasic changes in hit rates reported in prior studies come about, namely via changes in internal sensory response noise. However, the neural mechanisms corresponding to this change of internal response variability remain an important next step for further research. One possibility is that changes in sensory tuning occur as a consequence of local cortical inhibition, such as through lateral inhibition, as suggested by a recent study showing that alpha phase impacts a tilt illusion thought to depend on lateral inhibition in visual cortex (*Williams et al., 2024*). Another possibility is that feedforward inhibition such as that coming from rhythmic inhibitory activity in the thalamus (*Lorincz et al., 2009*) shapes the timing of stimulus-representing spiking activity in the visual cortex (*Dougherty et al., 2017*). A third possibility is that the phasic effect on sensitivity is partly a result of phasic top-down inputs that periodically provide disambiguating information to the perceptual decision process (*Sherman et al., 2016*; *Hetenyi et al., 2024*). While these accounts are not mutually exclusive, future research could attempt to disentangle these accounts by manipulating stimulus probability and using invasive recording techniques.

Our study demonstrates converging evidence that alpha-band phase is implicated in the fidelity of the internal visual representation, with less variable representations during one's optimal phase. Moreover, our results suggest that prior literature reporting phasic effects in the alpha-band range from both frontal and occipital regions may plausibly be reporting the same effect from a single projected dipole source; however, these results are also consistent with two synchronized alpha sources which are anti-phase. In sum, we show that alpha-band oscillations contribute to trial-to-trial changes in sensitivity through the variability of internal representations.

## Methods
### Participants

This study collected simultaneous EEG and behavioral reports from 6 human participants during a yes/no visual detection task (mean age: 28 years; 4/6 identified as female and the remaining identified as male; 3/6 reported being ethnically only White, 1/6 reported only Asian, 1/6 reported Asian and White, and 1/6 reported Asian [Indian] and Mexican). Each participant completed 43 task blocks (140 trials per block) across 4–5 EEG sessions and was compensated for their time ($100 total). Predetermined inclusion criteria included the presence of a discernable alpha peak in the power spectrum. No participants were excluded based on this criterion. All procedures performed in this study were reviewed and approved by the University of California, Santa Cruz Institutional Review Board. All

participants provided informed consent for the experiment, reported normal or corrected-to-normal vision, and were included in all analyses.

## Behavioral task

During the visual detection task, participants sat in a dimly lit room approximately 74 cm away from a gamma-corrected VIEWPixx EEG monitor (53.4 × 30 cm, 1920 × 1080 resolution, 120 Hz refresh rate) with their head stabilized on a chin rest. The experiment and all stimuli were coded in MATLAB (*The MathWorks Inc, 2022*) using Psychophysics Toolbox 3 (*Brainard, 1997*). Before beginning the main task blocks, participants completed one or more practice blocks with auditory feedback on incorrect trials, followed by a thresholding block which used a 1 up-2 down staircase to estimate a starting threshold. The starting threshold was computed by taking the average contrast level over the last 20 reversals. For the remainder of the study, the contrast was increased or decreased if performance on two consecutive task blocks fell below a d' of 1.2 or exceeded a d' of 1.8. Feedback was presented on the practice block(s) only.

Stimuli appeared on a uniformly gray background (approximately 50 cd/m²) with a small central fixation point that was physically isoluminant with the background. On each trial, a brief (8 ms; corresponding to a single screen refresh) patch appeared either to the right or left of fixation (3 degrees of visual angle [DVA] in diameter with a 5 DVA horizontal offset from center). On half of the trials (50%), the patch contained only filtered noise (stimulus-absent); on the remaining 50% of trials, the patch contained a vertical grating (two cycles per degree; randomized phase) embedded in noise (stimulus-present). On each trial, participants first provided a dual-response, which included whether they detected the grating (using either their right or left hand) and a concurrent confidence rating (using one of four fingers on the relevant hand). The hand used to report presence/absence was counterbalanced across sessions, whereas confidence was always reported on a 4-point scale (1 = 'guessing the patch was noise/a grating', 4 = 'certain the patch was noise/a grating') by using one of four fingers (1=pointer finger, 4=little finger). Next, participants reported the likelihood that they had imagined seeing either the noise or the grating through a yes/no response (results of the imagination question are not the main focus of this paper but can be found in the supplemental materials).

The noise was filtered between spatial frequencies (SF) 0.25–4 cpd with uniform power and had a root mean square (RMS) contrast of 70%. During stimulus-present trials, the contrast of the grating was initially informed by the thresholding block and then continuously titrated across blocks in order to maintain a d' between 1.2 and 1.8 (average grating contrast: 15%, SD: 1%). Throughout the block, each stimulus was identically presented twice using a double-pass procedure. This involved creating 70 unique stimuli per block (140 trials per block), which we then divided into two sequenced sets. Specifically, to account for fatigue effects within a block, the first set of 35 stimuli were presented on trials 1–35 and then repeated in the same order for trials 36–70. The second set was presented during trials 71–105 and then repeated in the same order for trials 106–140. Due to a coding error, the first three participants encountered significantly fewer double-pass presentations (M=596 pairs) than the last three participants (M=2,502 pairs). However, all participants were included in the response consistency analysis.

## EEG recording and data preprocessing

Continuous EEG data were acquired using Brain Vision Recorder with a 64-channel gel-based active electrode system (*actiCHamp Plus (64 channels), 2019*). Data preprocessing was performed using a custom MATLAB script (R2022a) in conjunction with the EEGLAB toolbox (*Delorme and Makeig, 2004*). Initial steps included applying a 0.1 Hz high-pass filter, downsampling to 500 Hz, and re-referencing to the median across all electrodes. The data were then segmented into 4-second epochs centered on stimulus onset and visually inspected for quality. Trials were excluded based on criteria such as excessive noise (e.g., voltage fluctuations >±150 µV), muscle activity (e.g., jaw or forehead movement, skin potentials, swallowing), or eye movements (e.g., blinks) occurring from –800–200 ms relative to stimulus onset. On average, 11% of trials were removed per session (range: 5–19%). Noisy channels were excluded and subsequently replaced using spherical interpolation (mean: 4.3 channels). Finally, the data were re-referenced to the average of all electrodes.

### Individual alpha frequency (IAF) and phase estimates

We computed each participant's IAF by performing a fast Fourier transform (FFT) on the Hamming tapered prestimulus window from –500–0 ms, which had been zero padded by a factor of 5 to increase frequency resolution. The data were then averaged across a subset of occipital-parietal electrodes (P1, Pz, P2, PO3, POz, PO4) and across trials, resulting in a single power spectrum. The IAF was the peak of that power spectrum in between 7 and 14 Hz (M=10.43; SD = 0.64) and was subsequently used to filter the single-trial data and choose the analysis electrodes.

Instantaneous phase estimates were computed using a Hilbert transform. We first prepared the raw data by applying a post-stimulus taper to the single-trial EEG data, which gradually attenuated the post-stimulus signal to zero by multiplying the data by a linearly spaced value from 1 (the original signal) to 0 across the 30–70 ms time span. This resulted in the post-stimulus data being flat after 70 ms, which is intended to minimize the evoked response in our data. Doing so allowed us to include the maximum amount of data in our peri-stimulus phase estimates while avoiding contamination from stimulus-evoked activity. We then filtered the tapered data by the IAF ± 2 Hz using a non-causal hamming-window sinc FIR filter and downsampled to 150 Hz.

For all analyses, except the frontal/occipital phase difference, we used the three posterior channels with the greatest alpha power in the IAF ± 2 Hz. This was individually determined by rank-ordering 17 of the posterior channels (Pz, P3, P7, O1, Oz, O2, P4, P8, P1, P5, PO7, PO3, POz, PO4, PO8, P6, and P2) and algorithmically choosing the three with the highest power. This ensured that electrode selection was made independent of performance and instead was based upon maximizing alpha signal strength. Each participant had a different combination of electrodes which were used in the analyses; however, the same three channels were used across sessions within a participant (participant 1: POz, PO3, O1; participant 2: P7, PO7, PO4; participant 3: P2, P1, Pz; participant 4: O1, Oz, O2; participant 5: O2, PO8, PO4; participant 6: Oz, O2, O1). Measurements for the phase-coupling, response consistency, and reverse correlation analyses were first computed on the single-channel level for each time point (–450–0 ms) and then averaged across these three channels and time.

To compute the phase differences (i.e., the d' phase angle between frontal/occipital channels and hits/false alarms), we took the circular difference of the two electrodes/conditions for each timepoint and channel and then circularly averaged these values. Significance was tested using a circular V-test, as implemented in the CircStats toolbox (*Berens, 2009*), which tests the null hypothesis that the data came from a uniform distribution around the unit circle against the alternative hypothesis that the distribution that is not uniformly distributed but has a mean of 180°.

### Signal detection theory (SDT)

To characterize whether changes in detection across the phase of alpha were due to changes in sensitivity or bias, we computed d' and c for each phase bin and for each electrode and time point.

D-prime (d') was calculated by subtracting the z-transformed false alarm rate (FAR) from the z-transformed hit rate (HR):

$$\mathbf{d'} = z\left(\mathrm{HR}\right) - z\left(\mathrm{FAR}\right)$$

Criterion (c) was computed by taking the sum of the z-transformed HR and FAR, then multiplying by negative one half:

$$\mathbf{c} = -1/2\left(z\left(\mathrm{HR}\right) + z\left(\mathrm{FAR}\right)\right)$$

A loglinear correction was applied to both HR and FAR to account for the assumption of a small number of false alarms or misses given an infinite number of trials (*Stanislaw and Todorov, 1999*).

### Phase–behavior coupling analysis

To quantify the phase-coupling effects, we transformed the SDT measurements into vectors which could be averaged to obtain their resultant vector. The direction of the vector, theta, indicated the phase bin (assigned as the center of the phase bin) and the length of the vector, rho, was the d' or c value of the given phase bin. The average of these vectors (i.e., the resultant vector) quantifies both the magnitude of a phase-coupling effect (i.e., the length) as well as directionality (i.e., the angle or optimal phase) for a given time point and channel. The length of the vector indicates the extent to

which behavior was modulated by a specific alpha phase and was the primary outcome of our analysis. In order to reduce sudden spikes in the resultant vector length time series, we applied a sliding average (in 50 ms time steps) using the MATLAB function (movmean.m).

To assess statistical significance, the results were analyzed using non-parametric cluster-based permutation tests in addition to binomial statistical testing. A permuted threshold with a corresponding alpha level of 0.05 was obtained by randomly shuffling the trial order, removing the relationship of the actual EEG data to the behavioral response while retaining the temporal structure of phase across the trial. After the real data was compared to the permuted threshold, we computed cluster correction by iteratively comparing each permuted dataset to the permutation threshold and obtaining the cluster sizes of temporally adjacent significant data from the permutations. The final significant cluster value corresponded to the 95th percentile of expected cluster sizes from chance alone. Permutated data were treated identically to the real data with the exception of shuffling the mapping between single trial EEG and behavior.

Additionally, we used a binomial probability testing framework that is designed for small sample sizes and treats each participant as an independent replication. As such, it computes the probability of having observed the number of statistically significant outcomes by chance given our sample size (*Schwarzkopf and Huang, 2024*). To assess significance at the participant level, we averaged the participant's resultant vector length and permutations from –450–0 ms and obtained the 95th percentile of the time-averaged permutations. We then compared the averaged resultant vector lengths to the permutation thresholds for each subject, which revealed three out of six significant subjects. We then used the MATLAB function myBinomTest.m (*Nelson, 2026*) to compute the p-value associated with the probability of having observed three out of six significant subjects by chance (with a false-positive rate of 0.05).

For the underlying d', c, HR, and FAR of a given time point at the group level, we shifted the individual participant data according to the highest d' value. Thus, the highest d' value was assigned to bin 8 and all other measurements were shifted by the same number of steps. Bin 8 was then omitted from further visualizations. The shifted data were then averaged across all time points from –450 ms to 0 ms, based on significant effects at the group level, and averaged across participants. No statistics were conducted on these shifted variables and instead are for visualization purposes only. To test whether the phase associated with the highest HR and FAR were counterphase (i.e., had a 180° phase opposition), we circularly averaged the resultant vector angle of hits and false alarms across the –450–0 ms time window for each channel. We then took the circular difference of the averaged hit and alarm angle and used a circular V-test (*Berens, 2009*) to compute whether the difference was significantly clustered around a mean of 180°.

## Response consistency

We used a double-pass presentation in order to observe the consistency of participants' responses to identical stimuli (see 'Behavioral task'). The logic of this manipulation is that any variability (i.e., inconsistency) in responses to repeated presentations of identical stimuli must be driven by internal, rather than external noise. Therefore, changes in response consistency provide a measure of internal noise. After identifying the trials which had an identical stimulus pair, we took the resultant vector angle of a given time point and channel and assigned each trial to either be in the participant's good phase, if it fell within ± 90° of the resultant vector angle, or bad phase, if it fell outside of these bounds. This resulted in the two trials either both having been presented during the participant's best phase, or with one or both of the trials falling within the bad phase. To quantify response consistency, we computed the proportion of trials in which participants provided the same response across the two identical trials. Since the optimal phase changes over time, the set of trials were classified as either both having occurred during the optimal phase (or otherwise) for each time point (from –450–0 ms) and channel. The proportion of consistent responses was then averaged across channels and time. A two-tailed paired-samples t-test was used to determine statistical significance.

## Reverse correlation

Our reverse correlation analysis follows closely from *Fernández et al., 2022*; *Xue et al., 2024* using code modified from the Carrasco lab (https://github.com/antoniofs23/reverse-correlation-demo; *Fernandez, 2022*). *Figure 3* illustrates the reverse correlation procedure which allowed us to quantify

the weighting of specific stimulus features in participants' detection judgments by discretizing the energy in SF and orientation for each stimulus and then regressing the single-trial energy (of a given SF and orientation) on the behavioral response (**Fernández et al., 2022**; **Xue et al., 2024**). To this end, we first obtained the single-trial energy profiles ($E_{\theta f}$) of the presented stimulus by convolving the image with a set of Gabor filters ($g_{\theta f}$) in quadrature phase, quantifying the amount of energy in the stimulus (S) for a given spatial frequency ($f$) and orientation ($\theta$). This procedure is expressed as:

$$E_{\theta f} = \sqrt{\left(S \cdot g_{\theta f sin}\right)^2 + \left(S \cdot g_{\theta f cos}\right)^2}$$

The filters were centered around the target features (SF = 2 cpd, orientation = 0°) and included 15 linear steps from 0.5 to 4 cpd in the spatial frequency dimension and 19 linear steps from –80–80° in the orientation dimension. In order to quantify the fluctuations in SF and orientation energy in the stimulus above and beyond the target energy we normalized each energy profile separately for target present and absent trials by subtracting the average across trials and dividing by the SD, allowing us to include all trials in our analysis (**Xue et al., 2024**).

Next, we regressed the single-trial energy of a given SF and orientation on the participant's detection response (285 total regressions) using a probit link function (i.e, the inverse cumulative normal distribution; $\varphi^{-1}$) to obtain a beta value indicating the relationship between the stimulus feature strength and the response across trials.

$$p(yes) = \varphi^{-1}\left(\beta_1 E_{\theta f} + \beta_0\right)$$

$\beta_0$ denotes the intercept and the regression slope $\beta_1$ reflects perceptual sensitivity to the corresponding orientation and SF. A larger positive $\beta_1$ value indicates greater sensitivity, whereas a value of 0 indicates no sensitivity. These $\beta_1$ values create the classification image (CI) which reveals sensory tuning towards information of stimulus relevant and irrelevant features. Lastly, the CI was averaged across positive and negative orientations of the same magnitude in order to decrease noise and aid subsequent model fitting.

To capture changes in perceptual tuning between the optimal and suboptimal phase, we implemented a model-based bootstrapping procedure (n=10,000) which compared the modulation of three parameters of a 2-D Gaussian fit to the CI using least squares.

$$Gauss = p_1 \cdot \exp\left(-\left[\frac{(x - p_2)^2}{2p_3^2} + \frac{(y - p_4)^2}{2p_5^2}\right]\right) + p_6$$

The initial Gaussian was fit to the optimal phase CI by allowing all six of the above parameters to vary, characterizing the mean of x ($p_2$) and y ($p_4$), the variance (SD) of x ($p_3$) and y ($p_5$), the offset ($p_6$), and the gain ($p_1$). The main parameters of interest for hypothesis testing are (1) the gain parameter, corresponding to a multiplicative increase in sensitivity to the target features akin to the multiplicative gain account in **Figures 1C and 2** the SD parameters, corresponding to the internal noise account in **Figures 1C and 3** the offset parameter, corresponding to a criterion shift account in **Figure 1C**. To focus on these three hypothesis-driven parameters, we used the initial six parameter fits from the optimal phase CI and then allowed the gain to be scaled by a multiplicative factor ('modulation factor'), the two SD parameters to be scaled by a common multiplicative factor (reducing the two SD effects to one common source of internal noise), and the offset parameter to be scaled. The remaining two parameters controlling the center of the 2-D Gaussian were allowed to vary as well but were not of interest for hypothesis testing. Thus, the modulation factor indicates how the initial parameters from the optimal phase fitting need to change, through a multiplicative factor, in order to account for the suboptimal phase data, with a modulation factor of 1 indicating no change and a modulation factor greater than 1 indicating an increase in that parameter for the suboptimal phase. We generated bootstrapped distributions of these modulation factors by fitting the group-averaged CI in the optimal and suboptimal phases 10,000 times, where each time the group average was based on a different random (with replacement) subset of participants. The resulting modulation factors for each of the three parameters were then compared to 1 using a two-sided bootstrapped p-value (p<0.05).

To complement the model fitting approach with a model-free comparison of the effect of alpha phase on the CI, we computed the difference CI between optimal and suboptimal phases; we

subtracted the suboptimal beta values from the optimal beta values for each bootstrapped iteration. In order to assess significance whether a particular SF and orientation was weighted differently between the optimal and suboptimal phase, we compared the difference beta values to 0 using a two-tailed bootstrapped p-value.

## Acknowledgements

We would like to sincerely thank Shutian Xue, Antonio Fernández, and Marisa Carrasco for their guidance on the stimulus design and analysis code. As well as thank Jacob Chaudhry, Vrishab Nukala, Aishwaroopa Narayanan, Emily Lincoln, Evan Yuan, Ari Chaw, Zoey Crisan, and Vidushi Singh for their help on this project.

## Additional information

### Funding
No external funding was received for this work.

### Author contributions
April Pilipenko, Conceptualization, Software, Formal analysis, Investigation, Visualization, Writing – original draft, Project administration; Alexandra McGowan, Investigation, Visualization; Jason Samaha, Conceptualization, Resources, Software, Formal analysis, Supervision, Methodology, Writing – review and editing

### Author ORCIDs
April Pilipenko ⬤ https://orcid.org/0009-0005-1084-229X
Jason Samaha ⬤ https://orcid.org/0000-0001-8010-5993

### Ethics
Human subjects: All procedures performed in this study were reviewed and approved by the University of California, Santa Cruz Institutional Review Board. All participants provided informed written consent prior to participation and were compensated for their time.

Reviewer #1 (Public review): https://doi.org/10.7554/eLife.110000.3.sa1
Reviewer #2 (Public review): https://doi.org/10.7554/eLife.110000.3.sa2
Author response https://doi.org/10.7554/eLife.110000.3.sa3

## Additional files

### Supplementary files
MDAR checklist

### Data availability
All electrophysiological and behavioral datasets, as well as the code for analysis, are freely available in Alpha-Band Phase Modulates Perceptual Sensitivity by Changing Internal Noise and Sensory Tuning at https://osf.io/eup3s.

The following dataset was generated:

| Author(s) | Year | Dataset title | Dataset URL | Database and Identifier |
|---|---|---|---|---|
| Pilipenko A, McGowan A, Samaha J | 2025 | Alpha-Band Phase Modulates Perceptual Sensitivity by Changing Internal Noise and Sensory Tuning | https://osf.io/eup3s | Open Science Framework, eup3s |

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
