## [Editor Report · eLife Assessment]

This **important** study explores how the phase of neural oscillations in the alpha band affects visual perception, indicating that perceptual performance varies due to changes in sensory precision rather than decision bias. The evidence is **convincing** in its experimental design and analytical approach. This work should interest cognitive neuroscientists who study perception and decision-making.

---

## [Referee Report · Reviewer #1 (Public review)]

[Editors' note: This version has been assessed by the Reviewing Editor without further input from the original reviewers. The authors have addressed the comments raised in the previous round of review.]

Summary:

In their paper entitled "Alpha-Band Phase Modulates Perceptual Sensitivity by Changing Internal Noise and Sensory Tuning," Pilipenko et al. investigate how pre-stimulus alpha phase influences near-threshold visual perception. The authors aim to clarify whether alpha phase primarily shifts the criterion, multiplicatively amplifies signals, or changes the effective variance and tuning of sensory evidence. Six observers completed many thousands of trials in a double-pass Gabor-in-noise detection task while an EEG was recorded. The authors combine signal detection theory, phase-resolved analyses, and reverse correlation to test mechanistic predictions. The experimental design and analysis pipeline provide a clear conceptual scaffold, with SDT-based schematic models that make the empirical results accessible even for readers who are not specialists in classification-image methods.

Strengths:

The study presents a coherent and well-executed investigation with several notable strengths. First, the main behavioral and EEG results in Figure 2 demonstrate robust pre-stimulus coupling between alpha phase and d′ across a substantial portion of the pre-stimulus interval, with little evidence that the criterion is modulated to a comparable extent. The inverse phasic relationship between hit and false-alarm rates maps clearly onto the variance-reduction account, and the response-consistency analysis offers an intuitive behavioral complement: when two identical stimuli are both presented at the participant's optimal phase, responses are more consistent than when one or both occur at suboptimal phases. The frontal-occipital phase-difference result suggests a coordinated rather than purely local phase mechanism, supporting the central claim that alpha phase is linked to changes in sensitivity that behave like changes in internal variability rather than simple gain or criterion shifts. Supplementary analyses showing that alpha power has only a limited relationship with d′ and confidence reassure readers that the main effects are genuinely phase-linked rather than a recasting of amplitude differences.

Second, the reverse-correlation results in Figure 3 extend this story in a satisfying way. The classification images and their Gaussian fits show that at the optimal phase, the weighting of stimulus energy is more sharply concentrated around target-relevant spatial frequencies and orientations, and the bootstrapped parameter distributions indicate that the suboptimal phase is best described by broader tuning and a modest change in gain rather than a pure criterion account. The authors' interpretation that optimal-phase perception reflects both reduced effective internal noise and sharpened sensory tuning is reasonable and well-supported. Overall, the data and figures largely achieve the stated aims, and the work is likely to have an impact both by clarifying the interpretation of alpha-phase effects and by illustrating a useful analytic framework that other groups can adopt.

---

## [Referee Report · Reviewer #2 (Public review)]

Summary:

The study of Pilipenko et al evaluated the role of alpha phase in a visual perception paradigm using the framework of signal detection theory and reverse correlation. Their findings suggest that phase-related modulations in perception are mediated by a reduction in internal noise and a moderate increase in tuning to relevant features of the stimuli in specific phases of the alpha cycle. Interestingly, the alpha phase did not affect the criterion. Criterion was related to modulations in alpha power, in agreement with previous research.

Strengths:

The experiment was carefully designed, and the analytical pipeline is original and suited to answer the research question. The authors frame the research question very well and propose several models that account for the possible mechanisms by which the alpha phase can modulate perception. This study can be very valuable for the ongoing discussion about the role of alpha activity in perception.

Conclusion:

This study addresses an important and timely question and proposes an original and well-thought-out analytical framework to investigate the role of alpha phase in visual perception. While the experimental design and theoretical motivation are strong, the very limited sample size substantially constrains the strength of the conclusions that can be drawn at the group level.

Bibliography:

Button, K., Ioannidis, J., Mokrysz, C. et al. Power failure: why small sample size undermines the reliability of neuroscience. Nat Rev Neurosci 14, 365-376 (2013). https://doi.org/10.1038/nrn3475

Tamar R Makin, Jean-Jacques Orban de Xivry (2019) Science Forum: Ten common statistical mistakes to watch out for when writing or reviewing a manuscript eLife 8:e48175 https://doi.org/10.7554/eLife.48175

---

## [Author Response]

The following is the authors’ response to the original reviews.

**Public Reviews:**

**Reviewer #1 (Public review):**
Summary:In their paper entitled "Alpha-Band Phase Modulates Perceptual Sensitivity by Changing Internal Noise and Sensory Tuning," Pilipenko et al. investigate how pre-stimulus alpha phase influences near-threshold visual perception. The authors aim to clarify whether alpha phase primarily shifts the criterion, multiplicatively amplifies signals, or changes the effective variance and tuning of sensory evidence. Six observers completed many thousands of trials in a double-pass Gabor-in-noise detection task while an EEG was recorded. The authors combine signal detection theory, phase-resolved analyses, and reverse correlation to test mechanistic predictions. The experimental design and analysis pipeline provide a clear conceptual scaffold, with SDT-based schematic models that make the empirical results accessible even for readers who are not specialists in classification-image methods.Strengths:The study presents a coherent and well-executed investigation with several notable strengths. First, the main behavioral and EEG results in Figure 2 demonstrate robust pre-stimulus coupling between alpha phase and d′ across a substantial portion of the pre-stimulus interval, with little evidence that the criterion is modulated to a comparable extent. The inverse phasic relationship between hit and false-alarm rates maps clearly onto the variance-reduction account, and the response-consistency analysis offers an intuitive behavioral complement: when two identical stimuli are both presented at the participant's optimal phase, responses are more consistent than when one or both occur at suboptimal phases. The frontal-occipital phase-difference result suggests a coordinated rather than purely local phase mechanism, supporting the central claim that alpha phase is linked to changes in sensitivity that behave like changes in internal variability rather than simple gain or criterion shifts. Supplementary analyses showing that alpha power has only a limited relationship with d′ and confidence reassure readers that the main effects are genuinely phase-linked rather than a recasting of amplitude differences.Second, the reverse-correlation results in Figure 3 extend this story in a satisfying way. The classification images and their Gaussian fits show that at the optimal phase, the weighting of stimulus energy is more sharply concentrated around target-relevant spatial frequencies and orientations, and the bootstrapped parameter distributions indicate that the suboptimal phase is best described by broader tuning and a modest change in gain rather than a pure criterion account. The authors' interpretation that optimal-phase perception reflects both reduced effective internal noise and sharpened sensory tuning is reasonable and well-supported. Overall, the data and figures largely achieve the stated aims, and the work is likely to have an impact both by clarifying the interpretation of alpha-phase effects and by illustrating a useful analytic framework that other groups can adopt.Weaknesses:The weaknesses are limited and relate primarily to framing and presentation rather than to the substance of the work. First, because contrast was titrated to maintain moderate performance (d′ between 1.2 and 1.8), the phase-linked changes in sensitivity appear modest in absolute terms, which could benefit from explicit contextualization. Second, a coding error resulted in unequal numbers of double-pass stimulus pairs across participants, which affects the interpretability of the response-consistency results. Third, several methodological details could be stated more explicitly to enhance transparency, including stimulus timing specifications, electrode selection criteria, and the purpose of phase alignment in group averaging. Finally, some mechanistic interpretations in the Discussion could be phrased more conservatively to clearly distinguish between measurement and inference, particularly regarding the relationship between reduced internal noise and sharpened tuning, and the physiological implementation of the frontal-occipital phase relationship.

We appreciate the reviewer’s thoughtful and constructive feedback, particularly regarding clarity and framing. In response, we have made several revisions to improve transparency and contextualization throughout the manuscript.

First, we now explicitly contextualize the relatively modest change in sensitivity by adding discussion of the contrast-titration procedure and its implications for effect size interpretation. Second, we address the coding error that led to unequal numbers of double-pass stimulus pairs across participants sooner in the manuscript by reporting the average number of pairs per participant in the Results (as well as the Methods), allowing for readers to interpret the results more appropriately. Third, we have provided additional detail, including precise stimulus timing parameters, electrode selection criteria, and a clearer explanation of the rationale for phase alignment in the Results (in addition to the Methods) section. Finally, we have revised portions of the Discussion to adopt more conservative language when interpreting our results, which more clearly distinguishes between empirical observations and mechanistic inferences, along with offering additional interpretations for the frontal-occipital phase relationship.

We believe these revisions substantially improve the clarity, transparency, and interpretability of the manuscript.

**Reviewer #2 (Public review):**
Summary:The study of Pilipenko et al evaluated the role of alpha phase in a visual perception paradigm using the framework of signal detection theory and reverse correlation. Their findings suggest that phase-related modulations in perception are mediated by a reduction in internal noise and a moderate increase in tuning to relevant features of the stimuli in specific phases of the alpha cycle. Interestingly, the alpha phase did not affect the criterion. Criterion was related to modulations in alpha power, in agreement with previous research.Strengths:The experiment was carefully designed, and the analytical pipeline is original and suited to answer the research question. The authors frame the research question very well and propose several models that account for the possible mechanisms by which the alpha phase can modulate perception. This study can be very valuable for the ongoing discussion about the role of alpha activity in perception.Weaknesses:The sample size collected (N = 6) is, in my opinion, too small for the statistical approach adopted (group level). It is well known that small sample sizes result in an increased likelihood of false positives; even in the case of true positives, effect sizes are inflated (Button et al., 2013; Tamar and Orban de Xivry, 2019), negatively affecting the replicability of the effect.Although the experimental design allows for an accurate characterization of the effects at the single-subject level, conclusions are drawn from group-level aggregated measures. With only six subjects, the estimation of between-subject variability is not reliable. The authors need to acknowledge that the sample size is too small; therefore, results should be interpreted with caution.Conclusion:This study addresses an important and timely question and proposes an original and well-thought-out analytical framework to investigate the role of alpha phase in visual perception. While the experimental design and theoretical motivation are strong, the very limited sample size substantially constrains the strength of the conclusions that can be drawn at the group level.Bibliography:Button, K., Ioannidis, J., Mokrysz, C. et al. Power failure: why small sample size undermines the reliability of neuroscience. Nat Rev Neurosci 14, 365-376 (2013). https://doi.org/10.1038/nrn3475Tamar R Makin, Jean-Jacques Orban de Xivry (2019) Science Forum: Ten common statistical mistakes to watch out for when writing or reviewing a manuscript eLife 8:e48175 https://doi.org/10.7554/eLife.48175

We thank the reviewer for their supportive remarks on our design and analysis, and for raising this important statistical concern about our sample size (n=6). Our choice of a small sample size was driven by methodological considerations. Specifically, our reverse correlation analysis requires a large number of trials per participant, as it estimates perceptual tuning by regressing behavioral responses against fluctuations in the energy of stimulus features (orientation and spatial frequency). This approach, as well as the computation of signal detection theory (SDT) metrics such as d′ and criterion, depends on high trial counts to obtain reliable estimates, particularly given that our analysis further subdivides trials across eight phase bins. For this reason, we prioritized collecting a large number of trials per participant (∼5,000), which is consistent with established practices in psychophysical research.

Importantly, our approach means that our design is reliable on the individual level, which motivated us to include a new binomial probability testing in our revised paper. This binomial test helps address concerns about the generalizability of our results. Binomial testing considers each participant as an independent replication of the effect and then computes the p-value associated with the probability of having observed the given number of statistically significant participants by chance, with a false positive rate of 0.05. In our data, 3 out of 6 participants showed significant effects, which corresponds to a probability of 0.002 of having observed these effects by chance alone. We believe this converging evidence supports the replicability and generalizability of our results. To improve the transparency of the single-subject data, we have included single-participant results in the Supplemental Materials to allow readers to directly assess the consistency of effects across individuals and to better contextualize between-subject variability.

Thank you again for your suggestions, we believe that these additions have greatly improved our manuscript by demonstrating the robustness of our findings and increasing the transparency of our single-subject results.

**Recommendations for the authors:**

**Reviewing Editor Comments:**
The issue of generalizability arose during the review process, as your results are based on a small sample of participants who undertook a very large number of trials. In the revised version, it would be useful to discuss why this approach is valid, especially in the context of linking EEG with modeling (i.e., why it is more powerful than having many participants with fewer trials), and the extent to which your results can generalize to the population.

We sincerely appreciate all of the helpful comments provided by the reviewers and hope we can address the concerns of our experimental approach. In the introduction, we have emphasized the importance of our current small sample size design, which allows us to reliably compute our signal detection theory metrics across 8 phase bins in addition to including the reverse correlation analysis. In the methods section, we have added a description of the binomial probability statistical framework, which addresses the generalizability of our results. In this framework, each participant is viewed as an independent replication and the p-value reflects the probability of having observed the number of individually significant subjects from the total sample size by chance. In this regard, observing a significant effect in 3 out of 6 participants (as in our study) from chance alone has a 0.002 probability, which we believe is unlikely and instead reflects a true effect present in the general population.

Below I have copied our changes in the introduction and methods sections.

“... in a large number of trials (6,020 per observer, n = 6) across multiple EEG sessions. This approach ensures a sufficient number of trials in order to reliably compute signal detection theory (SDT) metrics across multiple alpha phase bins while also affording enough statistical power for reverse correlation analysis (Xue et al., 2024), making it preferred over having a larger sample size with fewer trials.”

“Additionally, we used a binomial probability testing framework that is designed for small sample sizes and treats each participant as an independent replication. As such, it computes the probability of having observed the number of statistically significant outcomes by chance given our sample size (Schwarzkopf & Huang, 2024).”

**Reviewer #1 (Recommendations for the authors):**
My suggestions are intended to be light-touch and focused on strengthening the clarity and durability of the Reviewed Preprint rather than on additional experimentation or major new analyses.(1) Limitation statement for the double-pass coding error:Add a short statement in the Methods or Results acknowledging that the coding error led to markedly fewer repeated stimulus pairs for the first three participants than for the last three. For the response-consistency result in Figure 2E, a simple acknowledgement that the available evidence is stronger for some participants than others will help readers calibrate their confidence without detracting from the main story.

Thank you for this suggestion, we have now added a statement to this effect in the Results section, in addition to the description already mentioned in the Methods section.

“To examine this, we implemented a double-pass stimulus presentation (~600 stimulus pairs for participants 1-3 and ~2,500 pairs for participants 4-6) and analyzed participant’s response consistency (Xue et al., 2024) to two identical stimuli.”

(2) Contextualizing the titrated performance level:In the Discussion, explicitly note that contrast was titrated to keep d′ between approximately 1.2 and 1.8, which intentionally maintains moderate performance. This contextualization will help readers understand that while the phase-linked changes appear modest in absolute terms, they are mechanistically informative within this design.

Thank you, we have included a sentence to the Discussions speaking to this point.

“We also note that the observed modulation of d’ between optimal and suboptimal phases was relatively modest in absolute terms (0.21) in our study and could therefore require many trials per subject to detect. Two reasons for this modest effect size could be related to specific features of our task design. First, we titrated stimulus contrast to maintain consistent task performance. This titration could have reduced the magnitude of the phase effect on d’ that would otherwise be apparent if the stimulus intensity were kept constant. Additionally, the use of (relatively) high-contrast random noise likely means that trial-to-trial variability in perception is largely driven by random fluctuations in the noise properties and, to a lesser extent, internal brain state. Although both of these choices were necessary to perform SDT and reverse correlation analysis, they differ from many previous studies investigating alpha phase using only near-threshold detection in the absence of external noise and may contribute to an underestimation of the true effect size.”

(3) Methods clarifications:(a) Replace placeholder text such as "{plus minus}" and "{degree sign}" with the appropriate symbols, and ensure that any equations implied in the reverse-correlation section are fully present.

Thank you for bringing this to our attention, these placeholder texts are an artifact of the conversion process and we will correct this.

(b) State explicitly that the 8 ms stimulus duration corresponds to a single frame on your 120 Hz display, which will clarify the timing in Figure 1A and the pre-stimulus windows in the phase analyses.

Thank you, we have added language to both the Method and Results sections explicitly indicating that the 8 ms stimulus choice corresponds to a single screen refresh. Additionally, we changed the text in Figure 1A to include inter-trial interval timing (as opposed to merely saying “Start Trial”):

“(A) Task design. Each trial contained a brief, filtered-noise stimulus (8 ms; one screen refresh) presented to the right or left of fixation with equal probability.”

“Each participant (n = 6) completed 5-6 EEG sessions of a Yes/No detection paradigm whereby participants reported the presence or absence of a brief (8 ms; one screen refresh) vertical Gabor target (2 cycles per degree) with concurrent confidence judgments (see Figure 1A), along with an additional imagination judgement (reported in the supplemental materials).”

(c) In the description of the post-stimulus taper, consider phrasing the rationale in terms of minimizing contamination from evoked responses rather than asserting that the taper ends before the earliest evoked response, which keeps the argument correct without committing to a precise latency boundary.

Thank you for this suggestion. We have changed our rationale for the taper to “minimizing”, rather than avoiding, the evoked response.

“This resulted in the post-stimulus data being flat after 70 ms, which is intended to minimize the evoked response in our data.”

(4) Analysis transparency:(a) In the description of posterior electrode selection, explicitly note that channels were chosen solely on the basis of alpha power, independent of behavioural performance, and that the same electrodes were used for each participant across sessions.

We have gladly made this clarification to the methods.

“This was individually determined by rank-ordering 17 of the posterior channels (Pz, P3, P7, O1, Oz, O2, P4, P8, P1, P5, PO7, PO3, POz, PO4, PO8, P6, and P2) and algorithmically choosing the three with the highest power. This ensured that electrode selection was made independent of performance and instead was based upon maximizing alpha signal strength.”

(b) Describe the phase-alignment step used to center each participant's optimal bin before group averaging as a device for visualization and summary, and clarify that inferential statistics are based on the underlying, non-aligned data as appropriate. This will reassure readers who are cautious about circularity.

We agree that this should be made more explicit throughout the manuscript and have added statements clarifying this aspect in the Figure 2B caption, the Results, and Method sections.

“The data have been aligned across participants so that each individual's highest d’ was assigned to bin 8 (omitted from the plot), with the remaining data circularly shifted, and is averaged across -450 ms to stimulus onset. This graph is for visualization purposes only. Error bars represent ± 1 SEM. The pattern shows a clear phasic modulation of d’ across bins.”

“... requiring us to phase-align the performance data across participants in order to visualize the underlying phasic effects. To this end, we aligned all metrics (d’, c, HR, and FAR) by circularly shifting the data so that the bin with the highest d’ was assigned to bin 8, which was then omitted from further visualizations.”

“Bin 8 was then omitted from further visualizations. The shifted data were then averaged across all time points from -450 ms to 0 ms, based on significant effects at the group level, and averaged across participants. No statistics were conducted on these shifted variables and instead are for visualization purposes only.”

(c) Add a short note on the number of permutations and the cluster-forming threshold in the phase-coupling analyses, if not already stated in the Results or captions, to complete the description of your non-parametric testing procedure.

Thank you, we agree that reiterating this information in the Results section is helpful for the reader to clarify the analysis procedure.

“After smoothing the resultant vector length over time with a 50 ms moving average, we compared the observed vector lengths to a permuted threshold (95th percentile of 1,000 permutations) at each time point from –700 to 0 ms and performed cluster correction (95th percentile of the permuted cluster size) to account for multiple comparisons.”

(5) Discussion framing:Make one or two small adjustments to your mechanistic phrasing so that the distinctions between measurement and interpretation are fully explicit:(a) State that the combination of phase-d′ coupling, counterphased hit and false-alarm rates, response consistency, and phase-dependent classification images is "consistent with" a reduction in effective internal noise and sharper estimated tuning at optimal alpha phase, within the assumptions of your SDT and reverse-correlation framework.

Thank you for this suggestion. We have changed the language in the discussions to reflect this framing and interpretation of the results.

“Moreover, our data are consistent with a model in which the variability of internal responses changes systematically across the alpha cycle, as reflected in the inverse relationship between hit rate and false alarm rate.”

(b) Emphasize that reduced effective internal noise and sharpened sensory tuning are two complementary descriptions of a better match between sensory evidence and decision template rather than fully separable mechanisms.

Thank you, we have added this language for clarity of our interpretation.

“Together with decreases in the variance of sensory tuning during the optimal phase, our results suggest that alpha phase impacts sensitivity by shaping trial-to-trial variation in internal noise during perceptual decision making, leading to better matches between sensory evidence and decision templates as opposed to a change in the gain of internal sensory responses.”

(c) Note that the frontal-occipital phase relationship is consistent with a coordinated, possibly top-down component to the alpha-phase effect, while remaining agnostic about the precise physiological implementation.

Thank you for raising this additional interpretation. We have added this as a plausible alternative to the single-source account in the Discussion section.

“Moreover, our results suggest that prior literature reporting phasic effects in the alpha-band range from both frontal and occipital regions may plausibly be reporting the same effect from a single projected dipole source; however, these results are also consistent with two synchronized alpha sources which are anti-phase.”

**Reviewer #2 (Recommendations for the authors):**
Major issues:Given that collecting more data may not be doable, the authors should take some actions to test the reliability of their results. For instance, simulations could be run to test the robustness of the results with such a small sample size (Zoefel, 2019). It would also be of interest to include in the report statistics and plots at the individual level, not only the aggregates. It is also important to report which electrodes were used in the analysis for each of the subjects, in the Methods section, it is clearly stated that these electrodes differed between subjects.

Thank you for these suggestions. To assess the reliability of our results at the single-subject level, we have included a new binomial probability test which is a framework suitable for small sample size experiments with large trail numbers (Schwarzkopf & Huang, 2024). Binomial testing views each individual as an independent replication and considers the probability of having observed the number of significant participants given the total number tested participants, and outputs the probability of having observed the results by chance. We believe this framework adequately addresses the reviewer’s concern of generalizability in addition to being well-suited to the design of our study.

To assess individual significance, we averaged the resultant vector length and permutations over the analysis window from -450 to 0 ms. If the resultant vector length exceeded the permutation for that participant, then they were considered to be a significant participant. In total, 3 out of 6 participants (participants 1, 4, and 5) showed significant d’ coupling. The binomial probability (equivalent to a p-value) of having observed this outcome as a result of three false positives at the individual-subject level is very small (p = 0.002), which is sufficiently low for psychological studies.

Below is the text which we have added to the Results and Methods sections.

“To interrogate the robustness of our findings at the single-subject level, we adopted a test of binomial probability, which is a statistical framework that treats each individual as an independent replication and is ideal for small sample size studies that utilize a large number of trials per observer (Schwarzkopf & Huang, 2024). For our data, we assessed individual significance by averaging the actual and permuted resultant vector lengths across time (-450 to 0ms) and comparing the real vector length to the 95% percentile of the permuted datasets. With this approach, 3 out of 6 participants showed significant d’-phase coupling which corresponds to a binomial probability of p = 0.002, indicating a very low probability that we observed these results by chance alone.”

“Additionally, we used a binomial probability testing framework that is designed for small sample sizes and treats each participant as an independent replication. As such, it computes the probability of having observed the number of statistically significant outcomes by chance given our sample size (Schwarzkopf & Huang, 2024). To assess significance at the participant level, we averaged the participant’s resultant vector length and permutations from -450 to 0 ms and obtained the 95th percentile of the time-averaged permutations. We then compared the averaged resultant vector lengths to the permutation thresholds for each subject, which revealed 3 out of 6 significant subjects. We then used the MATLAB function myBinomTest.m (Nelson, 2026) to compute the p-value associated with the probability of having observed 3 out of 6 significant subjects by chance (with a false-positive rate of 0.05).”

To address the reviewer's second request, we now include a supplemental figure which has each individual’s results for the main analysis (see Supplementary figure 3). These graphs, in addition to the methods, now provides the reader with each participant’s given set of analysis electrodes.

“Each participant had a different combination of electrodes which were used in the analyses; however, the same three channels were used across sessions within a participant (participant 1: POz, PO3, O1; participant 2: P7, PO7, PO4; participant 3: P2, P1, Pz; participant 4: O1, Oz, O2; participant 5: O2, PO8, PO4; participant 6: Oz, O2, O1).”

As an alternative approach, linear mixed models (LMM) could be used for statistics, as they are more suitable for small sample sizes (Wiley et al., 2019). LMM improve generalization by modelling subject-specific random effects. Although raw circular data is not suitable for LMM, the sine and cosine of the phases could have been used as predictors, for instance. Given that data were collected for 6 different sessions, sessions could be included as a factor in the model to improve statistical power.

We appreciate the suggestion but feel that LMMs would be a challenge in this case not only because the main predictor variables are circular, but because the main outcome variables are not defined on the single-trial level and require many trials to be computed (e.g., classification images, SDT measures, response consistency). As such, computing these measures within a session may also lead to noisier estimates than we had designed our experiment for. We therefore prefer the more straightforward approach we have taken in the paper, which has now been supplemented by a binomial test of individual-subject level significance.

Given that the number of subjects is quite small, I believe that individual data should be presented (either in the main text or supplementary materials) also for figures: 2A, B, C and D.

Thank you, we have included all of these results to the individual graphs in the Supplemental Materials (see Supplementary figure 3).

In plot 2B (HR and FAR) a p-value = 0.015 appears. However, in the text you write:"Indeed, this showed that the difference between the HR and FAR vector angle was significantly clustered around a mean of 180{degree sign} (v = 3.78, p = 0.01), indicating that the phase angle associated with the greatest hits was counterphase to the phase angle associated with the greatest false alarms."Which one is correct? Or do they refer to different tests?

We appreciate you catching this confusing discrepancy. The two values refer to the same test which has a p-value of 0.0145. In the figure, this value was rounded to the thousandths decimal place (i.e., 0.015), whereas in the text it was rounded to the hundredths value (0.01). We now consistently report p-values out to three decimal places throughout the manuscript.

Did you perform any statistical test for phasic modulation of dprime and criterion? I say that because in Figure 2B, you state that the data shows a "clear phasic modulation of d' across bins", but no statistic is mentioned. On the other hand, in Figure 2D, you state, "We did not & observe any significant phase-dependent relationship between phase and criterion." Is this sentence referring to both 2C and 2D panels or only to 2C?

Figure 2B and 2D show the phase-behavior relationship across bins after aligning the phase bins to each participant's “best” d’ bin. This bin is omitted from the plots because it is used for alignment, making the analysis circular. Accordingly, these panels were intended purely for visualization and were not used for statistical inference. Additional language has been added to the figure caption highlighting this aspect.

“The data have been aligned across participants so that each individual's highest d’ was assigned to bin 8 (omitted from the plot), with the remaining data circularly shifted, and is averaged across -450 ms to stimulus onset. This graph is for visualization purposes only.”

The primary statistical test for phase-behavior coupling was performed using permutation testing of the resultant vector length, which quantifies the magnitude of phase-dependent modulation. These results are shown in Figures 2A (for d′) and 2C (for criterion). In the original manuscript, we reported only the time points that survived cluster-based correction, but did not explicitly report the cluster *p*-values. We have now added these cluster *p*-values to the manuscript for completeness.

“The data revealed significant cluster-corrected coupling between alpha phase and d’ in the prestimulus window from -220 ms until stimulus onset (cluster *p* = 0.046),...”

Additionally, we have changed the caption of Figure 2 to be separate for (C) and (D).

“(C) No evidence for the coupling of criterion to pre-stimulus alpha-band phase. Graph C reveals the time course of the resultant vector lengths for alpha phase-criterion coupling, which shows no significant phase-dependent relationship between phase and criterion.

(D) The underlying shifted c across phase bins (shifted to participants’ optimal phase, as in graph B) did not visually demonstrate a phasic modulation pattern.”

Minor issues:In general, the paper is very clear. I found a statement confusing in the Response consistency section:"To quantify response consistency, we computed the proportion of trials in which participants provided the same response across the two identical trials. This procedure was done for each channel at each time point (from -450 to 0 ms) and then averaged."Which makes no sense, as response consistency is independent of channel and time point. I believe here you refer to the phase, maybe by just changing the order (start with response consistency and then proceed to phase), the paragraph would be clearer.

We appreciate you catching this mistake. We have clarified the Methods section in the following way:

“To quantify response consistency, we computed the proportion of trials in which participants provided the same response across the two identical trials. Since the optimal phase changes over time, the set of trials were classified as either both having occurred during the optimal phase (or otherwise) for each time point (from -450 to 0 ms) and channel. The proportion of consistent responses was then averaged across channels and time.”

Could you include a plot of the power spectrum used for IAF estimation of all the subjects?

Thank you for the suggestion. In Supplemental Figure 3 we have included the power spectrum that was used to estimate IAF in addition to a topoplot of alpha power (IAF +/- 2 Hz) that has the analysis electrodes labelled.

Bibliography:Wiley RW, Rapp B. Statistical analysis in Small-N Designs: using linear mixed-effects modeling for evaluating intervention effectiveness. Aphasiology. 2019;33(1):1-30. doi: 10.1080/02687038.2018.1454884.Zoefel B, Davis MH, Valente G, Riecke L, How to test for phasic modulation of neural and behavioural responses, NeuroImage, Volume 202, 2019,116175, https://doi.org/10.1016/j.neuroimage.2019.116175.